**Data Availability Statement:** All data have been uploaded the Data Commons of the Pennsylvania State University and can be found through the DOI: https://doi.org/10.26208/bhzn-e084 and data

# Cover crop mixture expression is influenced by nitrogen availability and growing degree days

Barbara Baraibar[1☯]*, Ebony G. Murrell[2☯¤a], Brosi A. Bradley[2☯], Mary E. Barbercheck[3‡], David A. Mortensen[1¤b‡], Jason P. Kaye[2‡], Charles M. White[1☯]

1 Department of Plant Science, Penn State University, University Park, State College, Pennsylvania, United States of America, 2 Department of Ecosystem Science and Management, Penn State University, University Park, State College, Pennsylvania, United States of America, 3 Department of Entomology, Penn State University, University Park, State College, Pennsylvania, United States of America

☯ These authors contributed equally to this work.
¤a Current address: Crop Protection Ecology, The Land Institute, Salina, Kansas, United States of America
¤b Current address: Department of Agriculture, Nutrition and Foods Systems, University of New Hampshire, Durham, New Hampshire, United States of America
‡ These authors also contributed equally to this work.
* bbaraibar@hotmail.com

## Abstract

Cover crop mixtures can provide multiple ecosystem services but provisioning of these services is contingent upon the expression of component species in the mixture. From the same seed mixture, cover crop mixture expression varied greatly across farms and we hypothesized that this variation was correlated with soil inorganic nitrogen (N) concentrations and growing degree days. We measured fall and spring biomass of a standard five-species mixture of canola (*Brassica napus* L.), Austrian winter pea (*Pisum sativum* L), triticale (*x Triticosecale* Wittm.), red clover (*Trifolium pratense* L.) and crimson clover (*Trifolium incarnatum* L.) seeded at a research station and on 8 farms across Pennsylvania and New York in two consecutive years. At the research station, soil inorganic N (soil iN) availablity and cumulative fall growing degree days (GDD) were experimentally manipulated through fertilizer additions and planting date. Farmers seeded the standard mixture and a "farm-tuned" mixture of the same five species with component seeding rates adjusted to achieve farmer-desired services. We used Structural Equation Modeling to parse out the effects of soil iN and GDD on cover crop mixture expression. When soil iN and fall GDD were high, canola dominated the mixture, especially in the fall. Low soil iN favored legume species while a shorter growing season favored triticale. Changes in seeding rates influenced mixture composition in fall and spring but interacted with GDD to determine the final expression of the mixture. Our results show that when soil iN availability is high at the time of cover crop planting, highly competitive species can dominate mixtures which could potentially decrease services provided by other species, especially legumes. Early planting dates can exacerbate the dominance of aggressive species. Managers should choose cover crop species and seeding rates according to their soil iN and GDD to ensure the provision of desired services.

commons link: http://www.datacommons.psu.edu/commonswizard/MetadataDisplay.aspx?Dataset=6246 Excel files of the original data are currently available at this location. They can also be located by searching the lead author's name (Baraibar).

**Funding:** This work was supported by the USDA National Institute of Food and Agriculture Organic Research and Extension Initiative (https://nifa.usda.gov/funding-opportunity/organic-agriculture-research-and-extension-initiative) under Project PENW-2015-07433, Grant No. 2015-51300-24156, Accession No. 1007156 to JPK. The funders had no role in study design, data collection and analysis, decision to publish, or preparation of the manuscript.

**Competing interests:** The authors have declared that no competing interests exist.

## Introduction

Cover crops are unharvested crops planted in rotation between cash crops to provide ecosystem services, including reducing soil erosion, managing soil nutrients or suppressing weeds [1–3]. Interest in cover crops and in particular, cover crop mixtures is growing among farmers [4]. Cover crop mixtures comprised of multiple species can potentially provide a greater suite of benefits compared to cover crops grown as monocultures [5–9]. However, the expression of species in a mixture can differ across locations [5, 9, 10], thus making it difficult to predict the benefits of cover crop mixtures. Despite the increase in interest and adoption of cover crop mixtures, there is still little research examining what factors drive cover crop mixture expression. Such knowledge can help farmers make more informed decisions when combining cover crop species for particular goals.

Cover crop species in a mixture are usually chosen to provide specific services. High-yielding grass cover crops or brassicas are usually used to increase organic matter, decrease N leaching or suppress weeds [1, 3, 11], whereas legumes are mainly chosen to provide N for the following cash crop [5, 12]. Evaluations of the relationships between functional diversity and multifunctional provisioning of services by cover crops suggest that increased multifunctionality is associated with even representation, by biomass, of different cover crop species that possess divergent functional traits [2, 13].

To ensure that all component species in mixtures accumulate enough biomass to contribute to the desired services, farmers and researchers design mixtures based on seeding rates. Seeding rates of each species can be adjusted based on the number of species in the mixture and functional traits such as the competitiveness of each species, life form (annual, biennial) or cold tolerance [14]. However, cover crop biomass of each species in a mixture may not always reflect seeding rates. In one case, in mixtures with 90% legume seed by weight (and 10% grass), legume biomass was only 9.5% of total biomass at the end of the cover crop season [15]. Similarly, in a 25:75 cereal rye–hairy vetch (*Vicia villosa* Roth) mixture, cereal rye ended up being 43% of the cover crop stand composition, whereas vetch only accounted for 38% [16]. Thus, seeding rate alone is a poor predictor of final mixture expression because of the array of environmental and management filters that differentially affect cover crop species.

Timing of management [17, 18], species selection [17], and soil N availability [5, 6, 19] can interact with seeding rates to affect the resulting proportions of species in a mixture. In two- and three-species cover crop mixtures, high soil N availability decreased biomass of legumes in a mixture with non-legumes, likely because grasses and brassicas were more competitive as soil fertility increased [19–21, 5]. When supplemental N was added, mustards became highly dominant in two-species mixtures whereas lower soil N shifted the dominance from mustards to other species such as oats (*Avena sativa* L.) [22–24]. Conversely, legume species are more competitive in lower fertility soils because they can fix their own N [5, 15].

The length of the growing season, measured as growing degree days (GDD), can also filter or limit the growth of some species. Spring biomass of hairy vetch grown in a mixture with cereal rye decreased as total GDD decreased whereas biomass of cereal rye was not affected [25]. Similarly, a cover crop mixture composed of three, four and six species were heavily influenced by sowing date such that with low fall GDD, cereal rye completely dominated the mixtures [18].

While some research has been conducted to understand the underlying causes of variability in two species mixtures, there is little research on how more diverse cover crop mixtures respond to environmental variables across diverse farm settings. Furthermore, it is unclear whether farmers can manipulate mixture expression by adjusting seeding rates, or whether mixture expression is so influenced by environmental variables that farmers have little control

over it. To test the effect of N availability and fall and spring GDD on the expression of diverse cover crop mixtures, we conducted two experiments with the same five-species cover crop mixture (standard mixture). One experiment was located at the Russell E. Larson Agricultural Research Center at Rock Springs, PA (USA), and intentionally manipulated N availability and fall GDD with N fertilizer additions and planting dates, respectively. In the second experiment, eight farmer collaborators across Pennsylvania and New York planted the standard five species mixture plus a "farm-tuned" mixture at their farms. The "farm-tuned" mixture was composed of the same five species as the standard mixture but we adjusted the seeding rates to try to obtain a mixture that would meet each farmer's goals. All farms included in the study were USDA certified organic, grew grain crops and established the cover crop after a winter cereal and before maize (*Zea mays* L). Both experiments were repeated in two consecutive seasons. We hypothesized that: 1) Soil N at the time of cover crop seeding would affect the mixture composition such that legume species would accumulate more biomass where nitrogen availability was lower, and grasses and brassicas would dominate the mixtures where N availability was greater, 2) Season length in the fall would filter out some species and would interact with N availability to determine mixture expression, 3) Adjusting seeding rates (farm-tuning) would shift mixture expression and result in more biomass of the desired species. Our long-term goal is to enable farmers to use soil N and planting dates (GDD) to design cover crop mixtures and seed them at a rate that will provide even expression of species in the mixture, which will in turn maximize cover crop mixture multifuntionality.

## Materials and methods

### Research station experiment

**Site.** The experiment was conducted at the Pennsylvania State University's Russell E. Larson Agricultural Research Center, in Rock Springs, Pennsylvania, USA (40˚ 43' N, 77˚ 55' W, 350 m elevation). Annually this region receives 975 mm of precipitation with mean monthly temperatures ranging from 3˚C (January) to 22˚C (July). The dominant soil type at this location is a Hagerstown silt loam (fine, mixtured, semiactive, mesic Typic Hapludalf, [26]). Soil texture in our experimental field was predominantly clay loam with spatial variability in silt (range of 39.9–54.7%) and sand (14.0–27.0%) content across the field. The field had been managed according to organic regulations since 2003 and certified organic since 2006.

**Experimental design and sampling.** The experiment design was a split plot with four replicates with fall growing season length, approximated by three different planting dates, as the main effect and nitrogen availability as a nested treatment. Cover crop planting dates were chosen to represent a gradient of fall GDD (Table 1). In May 2016, the entire plot received 34

**Table 1. Cover crop mixture seeding dates and environmental conditions.**

| | 2016 | | | | 2017 | | | |
|---|---|---|---|---|---|---|---|---|
| Seeding date | Fall GDD | Total GDD | Soil iN | | Seeding date | Fall GDD | Total GDD | Soil iN | |
| | | | (+N) | (-N) | | | | (+N) | (-N) |
| Aug 4 | 1757 | 2254 | 79.2 | 44 | Aug 3 | 1535 | 1873 | 41.9 | 8 |
| Aug 24 | 1253 | 1750 | 56.4 | 29.1 | Aug 22 | 1160 | 1498 | 25.7 | 7.9 |
| Sep 7 | 948 | 1445 | 58.1 | 35 | Sep 12 | 826 | 1163 | 30.9 | 5.8 |

Fall GDD (growing degree days from planting to biomass sampling in November), and total GDD (from planting to termination) at Russell E. Larson Agricultural Research Center, Rocks Springs, PA. GDD in the spring were the same for all treatments within each year and extractable soil inorganic N ("Soil iN" in mg N $kg^{-1}$ soil) at cover crop planting on the treatments where we added N (+N) and those where we did not (-N) in 2016 and 2017.

Mg ha$^{-1}$ of dairy bedpack manure. In 2017, the experiment was replicated in an adjacent area and no manure was applied. In both years, sorghum-sudangrass (*Sorghum bicolor* (L.) Moench) was planted to the whole area in June and was left to grow until cover crop planting. A week prior to each cover crop planting date, the sorghum-sudangrass was mowed and incorporated with a rotary harrow. Fertilizer as 67.2 kg ha$^{-1}$ of N as sodium nitrate (Allganic Chilean Nitrate, 15-0-2) was applied to half of each plot, randomized within the planting date main plot. The soil was then prepared with a cultipacker and a mixture of canola (*Brassica napus* L., 2.8 kg ha$^{-1}$), Austrian winter pea (*Pisum sativum* L. ssp. *sativum* var. *arvense*, 15.9 kg ha$^{-1}$), red clover (*Trifolium pratense* L., 3 kg ha$^{-1}$), crimson clover (*Trifolium incarnatum* L., 8.4 kg ha$^{-1}$) and triticale (*x Triticosecale* Wittm. ex A. Camus, 24.9 kg ha$^{-1}$) was drilled with a cone seeder at a depth of 2.5 cm. Legumes were inoculated with a peat-based legume inoculant no more than 24 h prior to seeding. Plots were 6.7 m wide and 7.5 m long. Cover crop and weed biomass were sampled in November, before the first killing frosts, and again in late April, prior to cover crop termination, by clipping all plant material from two 0.25 m$^2$ quadrats. All plants were sorted by species, dried at 60ºC for one week and weighed. All cover crops in each year were terminated on the same date in spring (April 24 and 27 in 2017 and 2018, respectively).

In both years, prior to cover crop seeding and after Chilean nitrate application, extractable soil inorganic N ($NO_3^-$ -N plus $NH_4^+$ -N in units of mg N kg$^{-1}$ soil), which we refer to herein as "soil iN", was measured in each plot to a 20 cm depth by collecting 6 cores with a 0.02 m diameter soil probe and performing a KCl extraction on the soil samples [11]. Growing degree days from planting to fall biomass collection (fall GDD) and from 1 January to cover crop termination (spring GDD) were calculated using measurements from the National Oceanic and Atmospheric Administration [27].

**Statistical analysis.** Total cover crop biomass and biomass of each component species in the fall and spring were analyzed with linear mixtureed models in R (lmer, R Core Team 2018) with the categorical values of planting date (early, middle and late) and nitrogen availability (Chilean nitrate added or not), year and their interactions as fixed factors, and block as a random factor. Year was considered a fixed factor because of the application of manure prior to the start of the experiment in 2016 but not in 2017. Because year was highly significant, data were analyzed for each year separately. Data were log transformed when needed to ensure normality and homogeneity of variance. Cover crop mixture composition (kg ha$^{-1}$ of each component species) in the fall and in spring was analyzed with a permutation analysis of variance (PERMANOVA) in R (vegan package, R Core team 2018) to test the significance of the treatments (Soil iN and GDD).

## On-farm experiment

**Experimental design and sampling.** Participating farms were located across Pennsylvania (6 sites) and New York (two sites). We also planted treatment plots at the Rock Springs research station to represent a ninth organic "farm" site. The plots at the research station were located approximately one mile away from the ones used for the experiment described in the previous section. At all farms, plots containing the same five-species mixture used at the research station (standard) and a "farm-tuned" mixture were planted with a one-box drill in the summers of 2016 and 2017 following harvest of winter small grains. The seeding rates of the "farm-tuned" cover crop mixtures differed across farms (Table 2). Seeding rates were calculated based on a tool we developed to predict cover crop species' biomasses based on seeding rate, GDD, and soil tests from each farm; the seeding rates were then adjusted based on the services that each farmer prioritized from the mixture (S1 Appendix). All farmers decreased canola seeding rates compared to the standard mixture. Out of the 8 farmers, four plus the

**Table 2. Cover crop mixture seeding rates.**

| | Mixture | Farm-tuning | Pea | Canola | Crim. Clov | Red Clov. | Triticale |
|---|---|---|---|---|---|---|---|
| | | | | | kg/ha | | |
| | Standard | None | 15,9 | 2,8 | 8,39 | 3,0 | 24,9 |
| Farm 1 | Custom | L+T+ | 21,81 | 0,75 | 3,82 | 2,06 | 65,42 |
| Farm 2 | Custom | L+T+ | 14,54 | 0,75 | 7,63 | 4,12 | 32,71 |
| Farm 3 | Custom | L+T+ | 21,08 | 0,78 | 11,10 | 0,96 | 39,24 |
| Farm 4 | Custom | L+T+ | 16,72 | 0,75 | 11,07 | 1,37 | 43,18 |
| Farm 5 | Custom | L+T- | 36,35 | 0,75 | 7,63 | 2,75 | 6,54 |
| Farm 6 | Custom | L+T- | 24,65 | 1,21 | 8,85 | 2,21 | 10,51 |
| Farm 7 | Custom | L+T- | 29,08 | 0,75 | 15,27 | 0,69 | 13,08 |
| Farm 8 | Custom | L+T- | 23,35 | 0,94 | 13,62 | 0,74 | 12,84 |
| R. Station | Custom | L+T- | 11,67 | 0,71 | 2,89 | 6,31 | 19,55 |

All rates are kg/ha. The standard mixture was planted at all sites.L+T+ farms increased legume and triticale seeding rates compared to the standard mixture, L+T- farms increased legume but decreased triticale seeding rates compared to the standard mixture. All farms decreased canola seeding rates.

research station, increased legume seeding rates and decreased triticale (L+T- mixtures, hereafter), while four increased both triticale and legume seeding rates (L+T+ mixtures, hereafter) compared to the standard mixture (Table 2). Farmers who increased legume seeding rates and decreased triticale rates prioritized N provisioning for the following crop whereas farmers who increased both legume and triticale rates also sought weed suppression. At each farm, each mixture was sown in a randomized complete block design with four replications with plot sizes ranging from 800 to 2000 m$^2$. Some farmers applied manure before cover crop seeding whereas others did not. Extractable soil inorganic N ($NO_3^-$ N plus $NH_4^+$ N), hereafter "Soil iN" was measured at each farm prior to cover crop seeding, and growing degree days (GDD) and precipitation data were obtained via NOAA Climate Data Online [27] from the closest meteorological station to each farm (S2 Appendix).

Cover crop and weed biomass were sampled following the same protocols as at the research station. Logistical constraints limited sampling on the same date but it occurred as close to the first killing frosts in the fall and to cover crop termination in the spring as possible.

**Statistical analysis.** Data from the research station experiment and the on-farm sites were combined for analysis. To evaluate how soil iN at cover crop planting and fall and spring GDD influenced the species composition of the mixture, we conducted structural equation modelling (SEM) using the CALIS procedure in SAS 9.4 (SAS Institute, Cary, NC). SEMs translate hypothesized path diagrams into a set of structured linear equations that are solved simultaneously [28]. N availability (mg N kg$^{-1}$ soil) and GDD were included in the model as exogenous variables (Fig 1). The biomass of each component species (kg ha$^{-1}$) were included as endogenous variables, which can act as both predictor and response variables in an SEM [29]. Our full hypothesized SEM included direct effects of fall GDD on the fall and spring biomass of all component species, direct effects of spring GDD on spring biomass of all component species, and direct effects of soil iN on fall and spring biomass of the non-legume species triticale and canola. We also hypothesized that canola biomass in the fall would have a direct effect on the biomass of all other component species in the fall and spring. Finally, we hypothesized that the fall biomass of each species would have a direct effect on the spring biomass of that species. We used α = 0.05 to determine statistical significance of the path effects and added error covariance terms to the model as suggested by Lagrange multiplier tests provided by the modification option of the CALIS procedure.

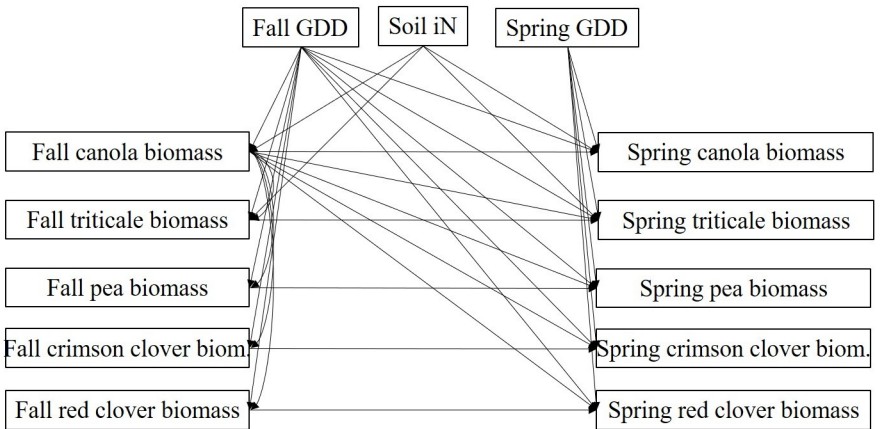

**Fig 1. Hypothesized path diagram to explain cover crop mixture expression.** GDD: growing degree days (base temp. 0C). Fall GDD are from cover crop planting to fall biomass sampling, spring GDD are from 1 January to spring sampling (close to cover crop termination). Soil iN: extractable soil $NO_3^-$ N and $NH_4^+$ N (mg kg$^{-1}$ soil), biomass (kg ha$^{-1}$).

The expression of "farm-tuned" mixtures was compared to the standard mixtures, using permutational multivariate analysis of variance (PERMANOVA) with type of mixture (standard, L+T- and L+T+) as a fixed factor and site-year as a random variable. Biomass of red clover, pea and crimson clover were summed up and analyzed as a single group called "legumes." Then, biomass of "farm-tuned" mixtures only was analyzed with PERMANOVA with soil iN and GDD (fall and spring) as fixed factors and site-year as a random variable. Significant factors and interactions were analysed with linear regressions using biomass of certain species in the mixture as response variables and environmental variables (GDD and soil iN) as explanatory variables. Finally, biomass of canola, triticale and legumes within each farm-tunning category (L+T+ and L+T-) was analyzed with linear regressions to test if changes in seeding rates resulted in the expected changes in biomass of these species. Analyses were performed in R (vegan and lme packages, R Core Team 2018).

## Results

### Fall

At the research station, the species expression of the cover crop mixtures responded to planting date in both years and to N availability in 2017 but not in 2016. In 2016, spring manure may have masked the effects of N fertilizer addition on mixture expression, as the treatment plots that did not receive Chilean nitrate had high soil iN levels at cover crop planting (Table 1). High soil iN levels caused canola to dominate the mixtures in both fertilized and unfertilized treatment plots, especially when seeded in early and middle August (Fig 2). Early-planted mixtures accumulated significantly more biomass than late-planted mixtures but the only difference in mixture composition across 2016 planting dates occurred for the September (late-) planted mixtures, which had a greater proportion of triticale than the early- and middle-planted mixtures. In 2017, N availability and planting date significantly influenced cover crop biomass but its effect varied across component species (Fig 2). Canola became dominant when planted early and when supplemental N was added. Triticale biomass was greater in earlier-2017 planted dates but it decreased with N addition. Nitrogen addition significantly decreased the biomass of winter pea and crimson clover, but only for the early and middle

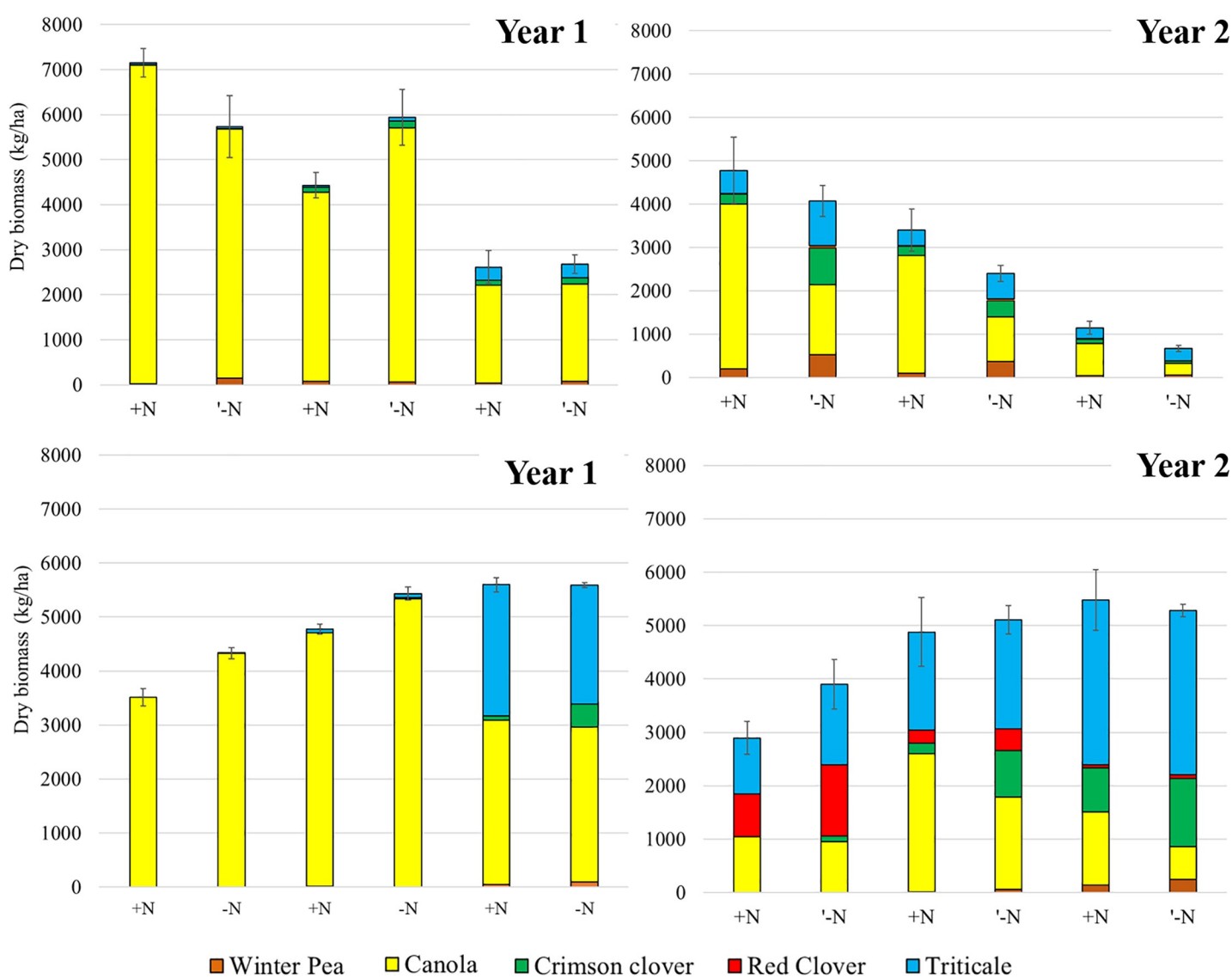

**Fig 2. Cover crop mixture biomass at the research station.** Biomass of each cover crop species at the research station site, as affected by planting date and N addition (+,-) in the fall of 2016 (year 1) and 2017 (year 2, top graphs) and spring of 2017 (year 1) and 2018 (year 2, bottom graphs). Standard error bars for the total biomass of each mixture are presented.

August 2017 planting dates. When the mixture was planted in September, the effect of N addition on mixture expression disappeared.

On-farm results were variable and mixture composition differed among farms and between years (Fig 3). We used a SEM to parse the effects of GDD and soil iN on biomass of each component cover crop species. Data from the research station and the on-farm experiment were combined in the SEM. The coefficients for each path in Fig 4 represent the expected change in cover crop biomass associated with a unit change in the predictor variable and only statistically significant paths are reported (P<0.05). Fit statistics for the full hypothetical path model, including error covariances, suggested a moderately good model fit (Chi-square = 93.8, df = 34, P<0.001; standardized root mean square residual = 0.07; Bentler's comparative fit index = 0.92). In the fitted path model, increasing GDD resulted in an increase of biomass of all cover crop species in the fall. However, the unstandardized path coefficients showed that

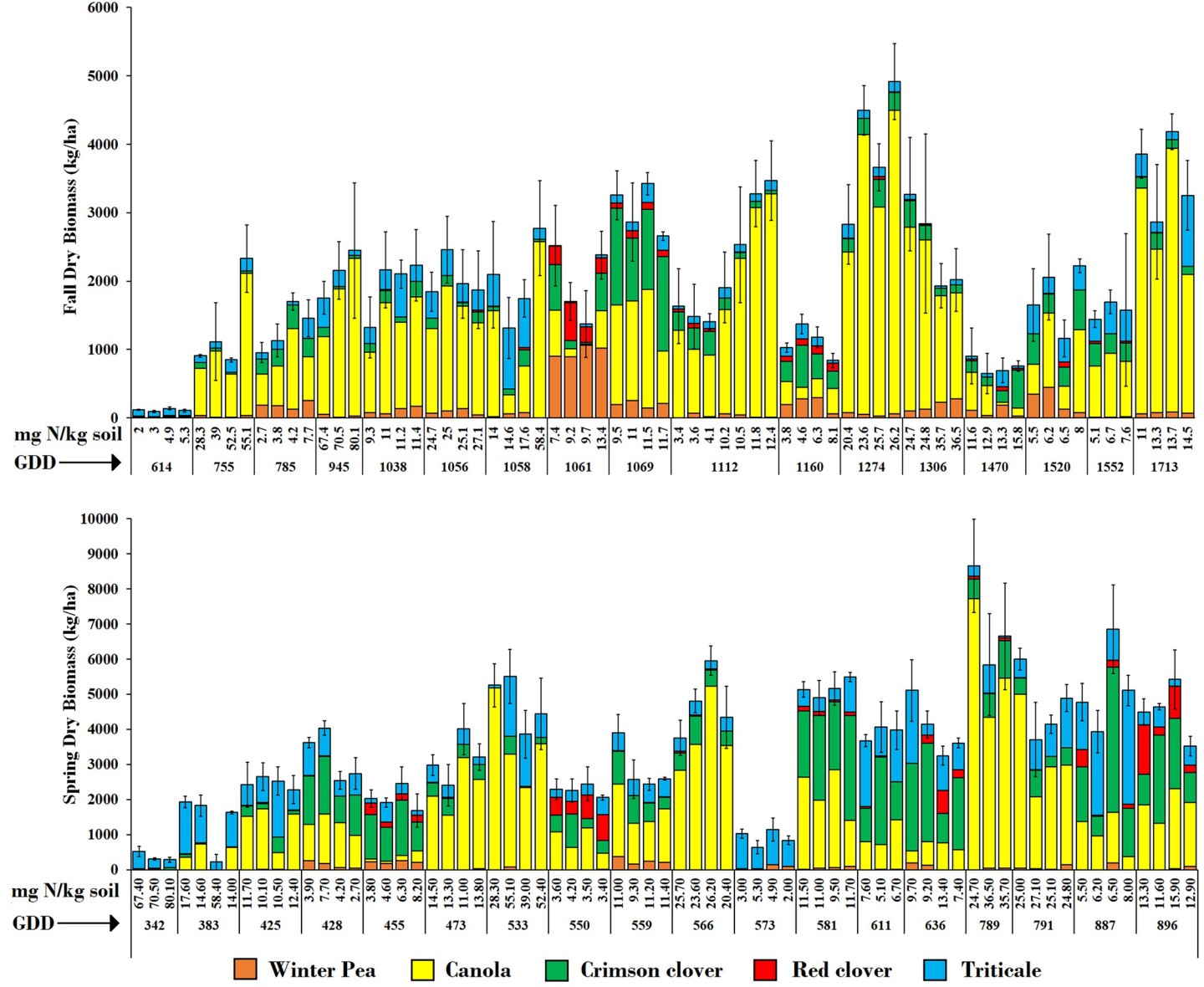

**Fig 3. Standard mixture fall and spring biomass.** Standard mixture dry biomass (kg ha⁻¹) of each cover crop species in the fall (top) and spring (bottom) at the on-farm sites in two years, as affected by N availability (mg N kg⁻¹ soil) and fall GDD (from planting to 31 December, top) and spring GDD (from 1 January to termination, bottom). The same mixture was planted at all locations. Each bar represents biomass expression in each of the four replicates of each site, and bars are sorted by increasing GDD and soil iN. Standard Errors for the total biomass are also included. Note that scales for the fall and spring are different.

for each additional fall GDD, canola biomass increased 2.8 kg ha⁻¹, whereas the increase was much lower for the other species (Fig 4). Soil iN had a strong direct effect on canola biomass; for each additional mg N kg⁻¹ soil, canola biomass increased 44 kg ha⁻¹. Canola biomass then had a direct effect of decreasing the biomass of all other cover crop species (negative unstandardized path coefficients from canola to other species, Fig 4). Even if soil iN did not have a significant direct effect on triticale, pea or the clover species, it had an indirect negative effect through increasing canola biomass. The negative effect of canola biomass on other species in fall also diminished the total effect of fall GDD on increasing biomass of the companion

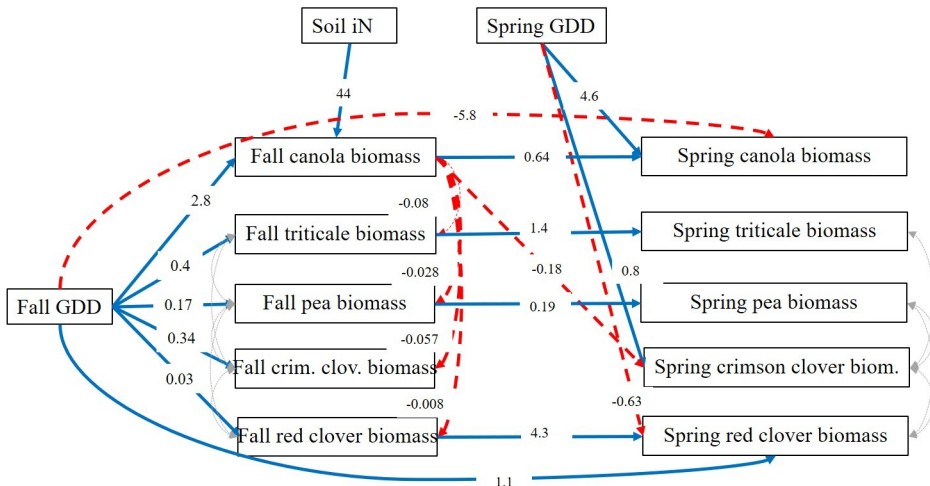

**Fig 4. Structural equation model results.** Significant unstandardized path coefficients from the Structural Equation Model (P<0.05). Blue solid arrows indicate a positive coefficient, red dashed arrows signify negative coefficients. Grey dotted arrows represent error covariance.

species, because canola biomass responded much more strongly to fall GDD than did the other species.

Adjusting seeding rates led to a decrease in canola biomass and an increase in legume biomass in all farm-tuned mixtures compared to the standard mixture. Triticale biomass increased in the L+T+ mixtures but it did not decrease as intended in the L+T- mixtures, compared to the standard mixture. Within the farm-tuned mixtures, farm-tuning and the interaction between soil iN and fall GDD significantly influenced mixture expression (Table 3, Fig 5). The effect of fall GDD on increasing the biomass of triticale, pea, crimson clover, and red clover was dimed by increasing soil iN, which stimulated canola growth and caused a negative feedback on the biomass of the companion species.

L+T+ and L+T-mixtures were significantly different but only differed in the biomass of triticale.

**Table 3. PERMANOVA analysis for farm-tuned cover crop mixtures.**

|  | Df | Sum of squares | F model | R2 | Pr (>F) |
|---|---|---|---|---|---|
| **Farm-tuning** | 1 | 1.18 | 13.35 | 0.11 | 0.02** |
| **Fall GDD** | 1 | 1.17 | 13.3 | 0.11 | 0.02* |
| **Soil iN** | 1 | 0.92 | 10.45 | 0.09 | 0.04 * |
| **Farm-tuning x Fall GDD** | 1 | 0.76 | 8.64 | 0.07 | 0.11 |
| **Farm-tuning x Soil iN** | 1 | 0.41 | 4.7 | 0.04 | 0.64 |
| **Fall GDD x Soil iN** | 1 | 0.59 | 6.73 | 0.05 | 0.004 ** |
| **Fall GDD x Farm-tuning x Soil iN** | 1 | 0.26 | 2.9 | 0.02 | 0.50 |
| **Residuals** | 64 | 5.64 |  | 0.52 |  |
| **Total** | 71 | 10.94 |  | 1.00 |  |

The influence of seeding rates (Farm-tuning: Increase in legume and triticale seeding rates (L+T+) or increase in legume and decrease in triticale seeding rates (L+T-)), fall GDD (from cover crop planting until fall biomass sampling) and soil iN ($NO_3$ and $NH_4$) on mixture composition in the fall. Data from the research station experiment and on-farm over two seasons are included. Site-year is included as a random factor.

Signif. codes: 0 '***' 0.001 '**' 0.01 '*' 0.05 '.' 0.1 ' ' 1.

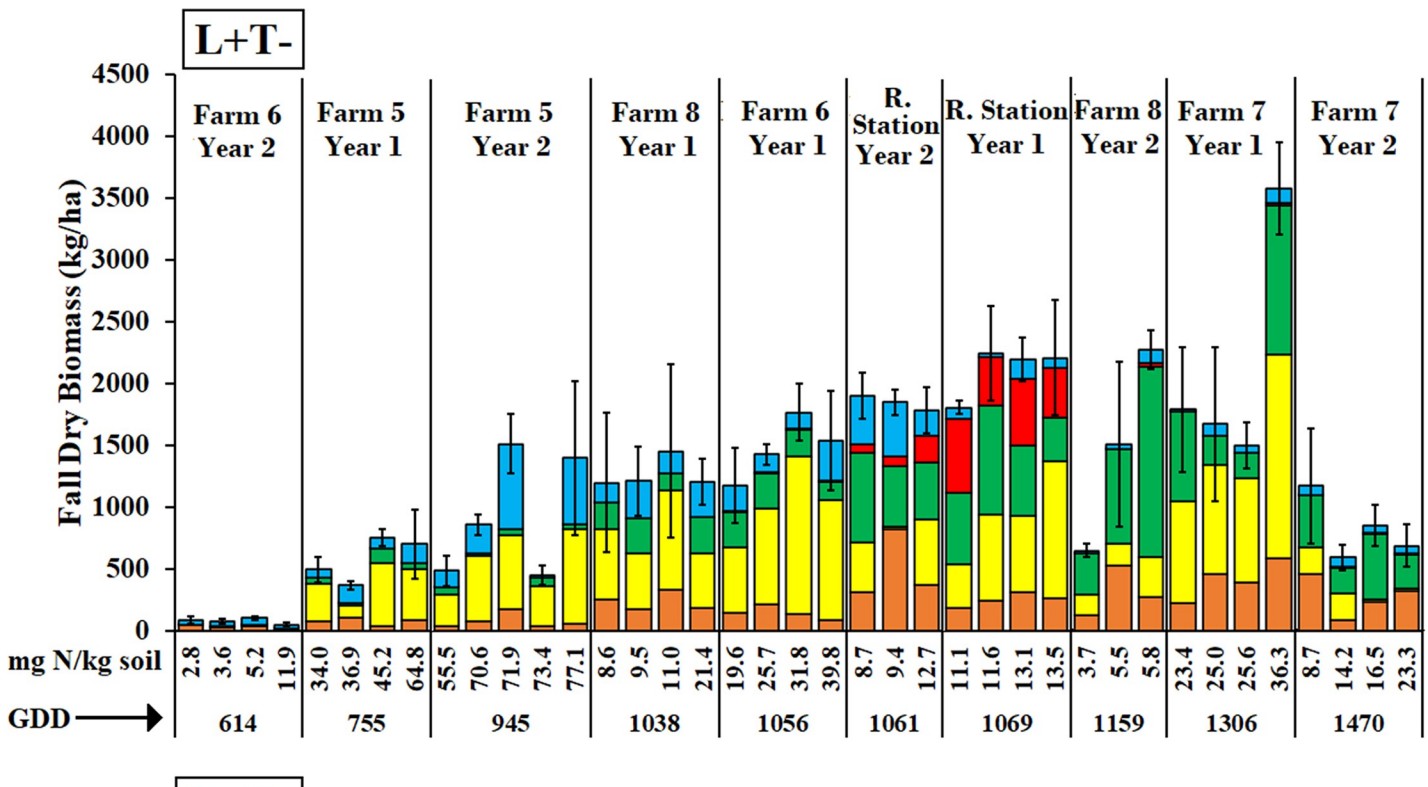

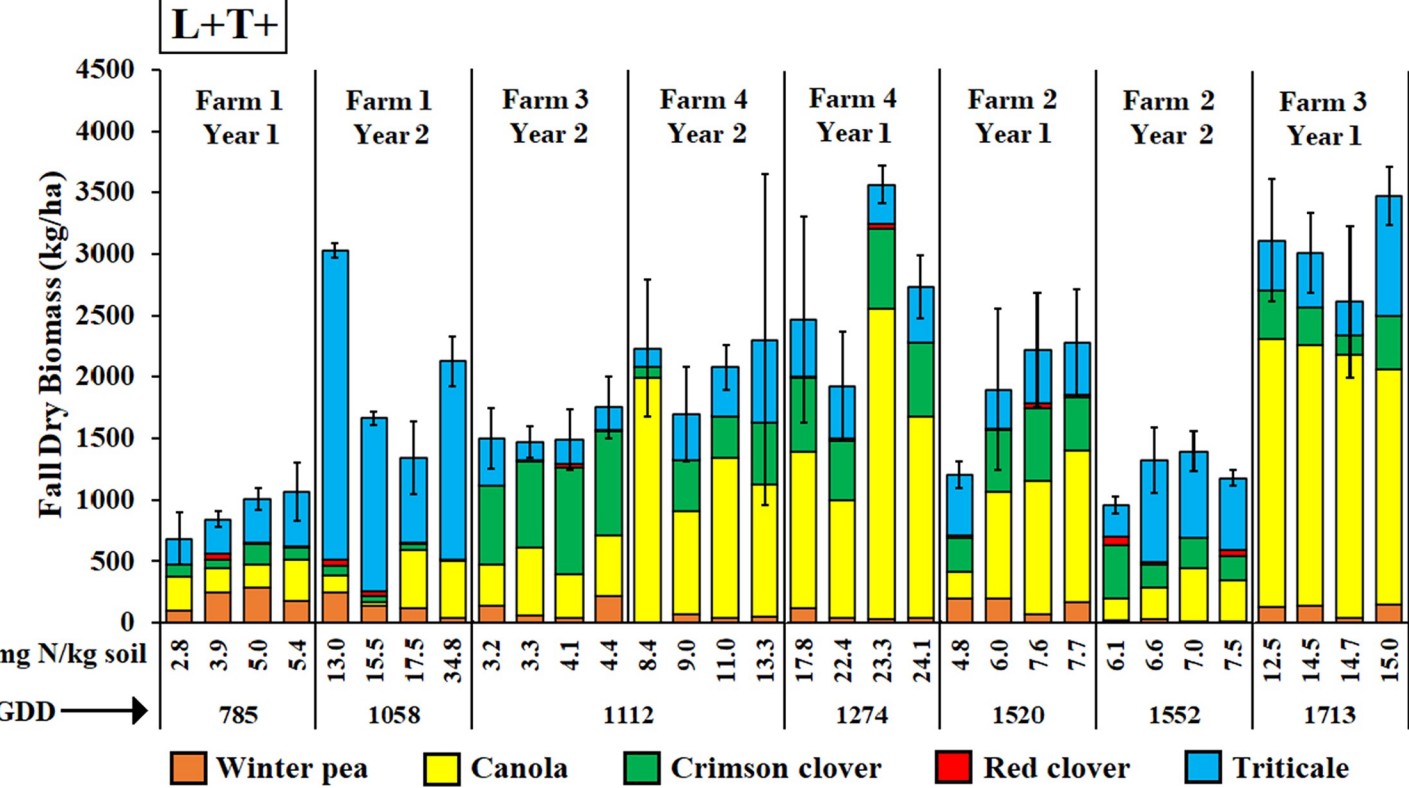

**Fig 5. Fall biomass of farm-tuned cover crop mixtures.** Dry fall biomass (kg ha$^{-1}$) of each cover crop species at farms that increased legume seeding rates but decreased triticale (L+T-, top) and at farms that increased both legume and triticale seeding rates (L+T+, bottom) in two years, as affected by fall GDD (from planting to fall biomass sampling) and soil inorganic N (in mg N kg$^{-1}$ soil). All farms decreased canola seeding rates (Table 2). Each bar represents biomass expression in each of the four replicates of each site, and bars are sorted by increasing GDD and soil iN. Standard Errors for the total biomass are also included.

## Spring

At the research farm, in the spring of both years, planting date significantly influenced cover crop biomass and composition but N addition did not (PERMANOVA, F = 18.9, p = 0.01 and F = 3.73, p = 0.07 respectively). In contrast to fall measures, the early-planted mixtures had lower biomass than those planted in mid-August and September (Fig 2). In spring 2017, fall canola dominance in the early- and middle-August planted mixtures continued until the spring, whereas mixtures planted in September had greater triticale biomass (Fig 2). In spring 2018, canola dominance was lower than in 2017 and its biomass was still significantly influenced by the addition of N in the fall ($F_{1,11}$ = 8.4, p = 0.01). The biomass of triticale increased in all treatments and was greater in the latest-planted mixtures, but N addition did not influence triticale biomass. For the legumes, while red clover flourished in the early-planting, crimson clover biomass was greater in the later-planted mixtures, and pea almost completely winter-killed in all mixtures. Biomass of both species of clover was greater in the treatments where no N had been added.

SEM results showed that the biomass of all component species in the spring was positively influenced by their biomass in the fall, except for crimson clover (Fig 4). Spring GDD positively influenced canola (for each GDD increase in spring, canola increased 4.6 kg ha$^{-1}$, Fig 4) and crimson clover (increased 0.8 kg ha$^{-1}$ for each spring GDD) and negatively influenced red clover (Fig 4). Interestingly, fall GDD negatively influenced spring canola and positively influenced red clover biomass in spring.

Composition of "farm-tuned" mixtures differed from the standard mixture but only in the biomass of triticale. Farm-tuning interacted with spring GDD such as that with a longer spring growing season, L+T- mixtures accumulated more canola and legume biomass, whereas GDD did not correlate with canola biomass in L+T+ mixtures. The interaction between soil iN and spring GDD also affected mixture composition (see PSU data repository https://doi.org/10.26208/bhzn-e084).

## Discussion

Our results confirm that soil iN and GDD are important drivers of cover crop mixture expression, especially in the fall, and that these factors interact with seeding rates to determine the resulting species composition that is expressed in a mixture. Our results were consistent with the hypotheses that legumes in a mixture accumulate more biomass when soil iN availability is low, and that brassicas accumulate more biomass when soil iN availability is high. However, the interaction between the different species and environmental conditions was nuanced. High soil iN did not result in the expected increase in triticale biomass. When soil iN was high, canola outcompeted all species in the mixture, including triticale. Competitive dominance of canola in the fall was exacerbated with longer fall growing season, even if increasing GDD favored all species. The reponse of canola to high soil iN availability was expected because of its high competitive ability to scavenge N and its leaf architecture, which quickly shades out other species [23, 24]. The inability of triticale to respond to greater N availability was likely related to its slower growth and preference for cooler temperatures compared to canola, as it is a winter annual that requires vernalization before proceeding in development. Indeed, the response of canola biomass to an increase in 1 fall GDD was 7 times greater than the response of triticale (Fig 4). With a shorter fall season (lower GDD), triticale was able to accumulate more biomass than when seeded earlier, probably because of less competition from canola, suggesting that delaying cover crop planting may ameliorate the dominance of highly competitive species like canola. Interestingly, canola biomass in the spring was negatively affected by high fall GDD. Larger canola plants are known to be more vulnerable to cold temperatures

[30] therefore, the winter-kill of some plants likely decreased canola spring biomass. These result suggest that early planting and high soil iN may not only disrupt the evenness of the cover crop biomass through excessive fall growth of canola, but also that winter-kill of canola may have negative effects on some of the services provided by the species in the spring, mainly flowering that can support pollinators and natural enemies, in addition to ground cover.

The response of legume biomass to soil iN has been reported for bicultures, as legumes tend to be less competitive in soils with greater availability of N [19, 5]. Our results show that this is also true for more complex mixtures and that even if soil iN influenced mixture composition mainly at the time of cover crop establishment, this effect carried through the winter. The length of the growing season also influenced the species of legumes that established. With high fall GDD, pea and crimson clover accumulated more biomass, while red clover accumulated very little. At the research station experiment, however, red clover was able to take advantage of the winter-kill of canola in the early-planted treatment and accumulated substantial biomass.

The large variability in the species expression of the standard cover crop mixture across farms challenges the common assumption that a mixture of the same cover crop species will express in the same way and provide the same services regardless of where it is seeded. This assumption may have economic implications because it underlies the practice of designing commercial seed mixtures with fixed seeding rates sold to farms across a wide variety of locations. Our results show that expression of diverse mixtures can vary depending on the climatic context and soil nutrient status and therefore, the identity and level of services derived from mixtures may also vary depending on farm context [5]. Some seed companies are already recognizing this context-dependent variability and are moving towards the design of custom mixtures. The variation in mixture composition between the fall and the spring also confirms that relative abundance of each species in a mixture can change over time, which could impact the services rendered by the mixture [18].

Adjusting seeding rates changed the biomass of component species but interacted with fall and spring GDD to determine mixture expression in both seasons. Also, soil iN and GDD interacted with each other (independently of farm-tuning) to influence mixture expression. Despite large variability across farms, decreasing canola seeding rates resulted in lower biomass in the farm-tuned mixtures compared to the standard mixture. Lower canola biomass probably freed space and resources for all other species to grow and allowed legumes to accumulate more biomass. However, the kind of "farm-tuning" also influenced mixture composition. Canola biomass was greater in the L+T+ mixtures than in the L+T- mixtures even though all farms decreased canola seeding rates in a similar way. L+T+ farms had a slightly longer growing season than L+T- farms, which could have allowed canola to accumulate more biomass in the L+T+ farms. These results suggest that environmental factors can be more important than seeding rates in determining mixture expression and that mixtures need to be "farm-tuned" to each set of conditions. Depending on the farm, changing species included in the mixture may have greater impact on resulting biomass than merely adjusting the seeding rate. One area worthy of future research is to determine if some cover crop species, such as canola, act as a keystone species whose presence regulates the community composition and ecosystem services provided by the cover crop mixture. In some cases, a keystone species may even serve to disrupt the overall functioning of the cover crop mixture, or introduce such a high level of sensitivity to environmental conditions that it reduces the stability of services provided by the cover crop over time and space, such that it should be avoided in cover crop mixtures altogether. This avenue of research is particularly needed because most farmers who used cover crops mixtures reported that they design their own mixtures [4].

The results of this study have further practical implications for management of cover crop mixtures. Farmers who apply fertilizers and animal manures in the fall, and thus, increase soil iN availability at the time of planting, should carefully select species and seeding rates to prevent dominant species from outcompeting all others. For the same reason, farmers may consider changing the time of fertilizer application to the spring or delaying cover crop seeding to later in the fall. Similarly, with high available soil iN, legumes are not expected to accumulate abundant biomass and therefore, farmers must decide whether it is worth including them in the mixture.

Farms in this study encompassed a wide range of geographical locations, soil types and management practices. That both soil iN and GDD significantly explained species expression of the standard mixture across farms suggests that they are critical factors determining competitive interactions among cover crop species. Cover crop mixtures are being increasingly adopted by farmers to provide a range of ecosystem services [4]. Realization of most of those services will depend on the amount of biomass of each species or functional group in the mixture. Farm-tuning should go beyond adjusting seeding rates and must encompass tailoring cover crop mixtures to each particular farm environment, soil iN, climate, field, and set of goals. This approach is more likely to provide the desired benefits to farmers than standardized, non-locally-adapted mixtures.

## Supporting information

**S1 Appendix. Farm-tunned mixtures design procedure.**
(DOCX)

**S2 Appendix. Cover crop mixture management and environmental conditions at the on-farm experimental locations.**
(DOCX)

## Acknowledgments

We are grateful to the staff of the Russell E. Larson Agricultural Research Center, and especially to Dayton Spackman, for planting, managing, and assisting in data collection in our experimental plots, and to many undergraduate assistants for their assistance with data collection. We thank farmers Ben Dobson, Gary Hoover, Bob and Jonathan Keller, Steve Misera, Thor Oechsner, Michael Ranck, Glenn Rex, and Bucky Ziegler for their participation and for planting and managing the cover crop mixtures. Finally we thank PSU educators David Hartman, David Wilson, Liz Bosak, Anna Busch, Rachel Milliron, and former educator Mena Hautau, and Cornell Extension educators Janice Degni, Christian Malszaski and former educator Justin O'Dea for their help with the research.

## Author Contributions

**Conceptualization:** Barbara Baraibar, Ebony G. Murrell, Brosi A. Bradley, Mary E. Barbercheck, David A. Mortensen, Jason P. Kaye, Charles M. White.

**Formal analysis:** Barbara Baraibar, Charles M. White.

**Investigation:** Barbara Baraibar, Ebony G. Murrell, Brosi A. Bradley, Charles M. White.

**Writing – original draft:** Barbara Baraibar.

**Writing – review & editing:** Ebony G. Murrell, Brosi A. Bradley, Mary E. Barbercheck, David A. Mortensen, Jason P. Kaye, Charles M. White.

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
