## [Decision Letter · Decision Letter 0]

4 Jun 2020

PONE-D-20-11469

Cover crop mixture expression is influenced by nitrogen availability and growing degree days

PLOS ONE

Dear Dr. Baraibar,

Thank you for submitting your manuscript to PLOS ONE. After careful consideration, we feel that it has merit but does not fully meet PLOS ONE’s publication criteria as it currently stands. Therefore, we invite you to submit a revised version of the manuscript that addresses the points raised during the review process.

We look forward to receiving your revised manuscript.

Kind regards,

Pirjo Mäkelä, DSc

Academic Editor

PLOS ONE

2. We note that you have included the phrase “data not shown” in your manuscript. Unfortunately, this does not meet our data sharing requirements. PLOS does not permit references to inaccessible data. We require that authors provide all relevant data within the paper, Supporting Information files, or in an acceptable, public repository. Please add a citation to support this phrase or upload the data that corresponds with these findings to a stable repository (such as Figshare or Dryad) and provide and URLs, DOIs, or accession numbers that may be used to access these data. Or, if the data are not a core part of the research being presented in your study, we ask that you remove the phrase that refers to these data

Reviewers' comments:

Reviewer's Responses to Questions

**Comments to the Author**

1. Is the manuscript technically sound, and do the data support the conclusions?

Reviewer #1: Yes

Reviewer #2: Yes

2. Has the statistical analysis been performed appropriately and rigorously? 

Reviewer #1: Yes

Reviewer #2: Yes

3. Have the authors made all data underlying the findings in their manuscript fully available?

Reviewer #1: No

Reviewer #2: Yes

4. Is the manuscript presented in an intelligible fashion and written in standard English?

Reviewer #1: Yes

Reviewer #2: Yes

5. Review Comments to the Author

Reviewer #1: Dear authors,

It was a pleasure to read your original manuscript and excellent that your research results are very practical for real farms. I have only minor suggestions:

1. Please use the WRB soil classification, or if you keep the soil classification as it is written now, please add in parenthesis what classification are you using

2. In tableA appendix 2 please add more information about the farms: approximate coordinates, min/max temperature of growing season, accumulated rainfall, and soil type

3. Please check the font size of the x axis and resolution of figures, in the .pdf the resolution is rather low

4. Figures 3 and 5 would benefit from adding S.E. bars

5. The text says that the interaction of soil inorganic Nitrogen and spring GDD affected the mixture composition but this data is not shown. Please justify why did you took this approach? wouldn't it be possible to make the data available for supplementary information?

Reviewer #2: The manuscript “Cover crop mixture expression is influenced by nitrogen availability and growing degree days” is a research paper investigating the effect of nitrogen availability and length of growing season (expressed as growing degree days, GDD) on the composition of cover crop mixtures seeded in 9 locations (research station and on-farm fields) in two consecutive years. Cover crop species composition was determined by sampling aboveground biomass in fall and spring. Overall, the results show that seeding rates of different species did not determine the biomass composition of the cover crop mixture per se, but nitrogen availability and GDD influenced the expression. This is an important and timely finding, which contributes to advancing the current understanding of cover crop mixtures for the provision of multiple ecosystem services, which depend on the actual expression of the mixture. The manuscript is very well developed, and I did not find any major flaws in the presented study. Therefore, I would recommend publication after few minor revisions, which I have listed below.

Consider using keywords that are not already in the title, to increase the chances of your manuscript to be found.

Line 125: You write that you used a split-split plot design, but I can only see two factors (GDD as main and N availability as nested). Should it be split-plot?

Line 182: Based on the current phrasing, it is not clear if the research station trial is the same as the one described in the previous section or another one. Could you clarify it in the text?

Line 195: Here you write that there are four replicates in each farm, but in Figures 3 and 5 you show data from only one plot per farm. I don´t follow the reasoning for this choice, which seems counter-productive to me. Could you explain?

Line 302: In the text, could you briefly explain the interaction, as you did in lines 342-344?

6. PLOS authors have the option to publish the peer review history of their article (what does this mean?). If published, this will include your full peer review and any attached files.

Reviewer #1: No

Reviewer #2: No

---

## [Author Response · Author response to Decision Letter 0]

22 Jun 2020

Editor

 We have modified the manuscript to meet PLOS ONE’s style requirements.

2. We note that you have included the phrase “data not shown” in your manuscript. Unfortunately, this does not meet our data sharing requirements. PLOS does not permit references to inaccessible data. We require that authors provide all relevant data within the paper, Supporting Information files, or in an acceptable, public repository. Please add a citation to support this phrase or upload the data that corresponds with these findings to a stable repository (such as Figshare or Dryad) and provide and URLs, DOIs, or accession numbers that may be used to access these data. Or, if the data are not a core part of the research being presented in your study, we ask that you remove the phrase that refers to these data

 We have added a citation to the repository where the data can be found. 

https://doi.org/10.26208/bhzn-e084 and http://www.datacommons.psu.edu/commonswizard/MetadataDisplay.aspx?Dataset=6246

Yes. Data has been uploaded to the Penn State Data Repository called Data Commons https://doi.org/10.26208/bhzn-e084. We have added the DOI number to the new Data Availability statement.

Reviewer 1

1. Please use the WRB soil classification, or if you keep the soil classification as it is written now, please add in parenthesis what classification are you using

We have added the reference to the classification we are using which is the Soil Survey Staff, 1999. We also have added it to the reference list and changed the numbers of the reference list to include this one.

2. In tableA appendix 2 please add more information about the farms: approximate coordinates, min/max temperature of growing season, accumulated rainfall, and soil type

We have added the requested information. We did not include coordinates to preserve the privacy of our on/farm collaborators but we did include approximate latitude and longitude, state and county where the farm was located so readers can have an idea of the location.

3. Please check the font size of the x axis and resolution of figures, in the .pdf the resolution is rather low

We have improved the quality of figures 3 and 5 so the x axes are more readable. We have also adjusted the colors so they are easier to distinguish for color/blinded readers.

4. Figures 3 and 5 would benefit from adding S.E. bars

We have added S.E. bars to both figures. We have added a line to the figure caption to reflect this change.

5. The text says that the interaction of soil inorganic Nitrogen and spring GDD affected the mixture composition but this data is not shown. Please justify why did you took this approach? wouldn't it be possible to make the data available for supplementary information?

We have added the sentence that the data are available at the PSU Data Commons Repository. Data can be accessed here: https://doi.org/10.26208/bhzn-e084

6. Line 422. Comment: if you have enough data on soil type for each farm, it would be worth to discuss if there were any trends regarding soil type effect on the cover crop mixtures establishment and biomass

That would be really interesting but the reported soil series was derived from a soil map and the hypothesis for the study was about soil iN, therefore we did not collect ancillary data related to the soil type and cannot provide a broader analysis.

Reviewer 2

1. Consider using keywords that are not already in the title, to increase the chances of your manuscript to be found.

We have deleted the keyword “cover crops” and added the ones: canola, legumes, triticale, clover, Austrian winter pea, seeding rates 

2. Line 125: You write that you used a split-split plot design, but I can only see two factors (GDD as main and N availability as nested). Should it be split-plot?

Yes, thank you. We have corrected it.

3. Line 182: Based on the current phrasing, it is not clear if the research station trial is the same as the one described in the previous section or another one. Could you clarify it in the text?

Yes. We have added the line: “The plots at the research station were located approximately one mile away from the ones used for the experiment described in the previous section”

4. Line 195: Here you write that there are four replicates in each farm, but in Figures 3 and 5 you show data from only one plot per farm. I don´t follow the reasoning for this choice, which seems counter-productive to me. Could you explain?

We have realized that captions for figures 3 and 5 were not correct when they stated that “Each bar represents biomass expression in a single replicate of each site”. In fact, each of the four bars in figures 3 and 5 represent the biomass in each of the four replicates in each site. 

We have corrected the captions to reflect this.

5. Line 302: In the text, could you briefly explain the interaction, as you did in lines 342-344?

We realized there had been an error entering the data on this table and the interaction between farm-tuning and fall GDD was not significant. We have corrected that. Farm-tuning alone did influence mixture composition. Also, the interaction between soil iN and fall GDD was significant. 

We have added a sentence to explain this interaction: “The effect of fall GDD on increasing the biomass of triticale, pea, crimson clover, and red clover was dimed by increasing soil iN, which stimulated canola growth and caused a negative feedback on the biomass of the companion species”.

---

## [Editor Report · Decision Letter 1]

24 Jun 2020

Cover crop mixture expression is influenced by nitrogen availability and growing degree days

PONE-D-20-11469R1

Dear Dr. Baraibar,

We’re pleased to inform you that your manuscript has been judged scientifically suitable for publication and will be formally accepted for publication once it meets all outstanding technical requirements.

Kind regards,

Pirjo Mäkelä, DSc

Academic Editor

PLOS ONE
---

## [Editor Report · Acceptance letter]

14 Jul 2020

PONE-D-20-11469R1 

Cover crop mixture expression is influenced by nitrogen availability and growing degree days 

Dear Dr. Baraibar:

I'm pleased to inform you that your manuscript has been deemed suitable for publication in PLOS ONE. Congratulations! Your manuscript is now with our production department. 

Kind regards, 

on behalf of

Prof. Pirjo Mäkelä 

Academic Editor

PLOS ONE